# Visual preference for social vs. non-social images in young children with autism spectrum disorders. An eye tracking study

Julia Vacas[1,2,3]*, Adoración Antolí[1,2,3,4], Araceli Sánchez-Raya[1,2,3,4], Carolina Pérez-Dueñas[1,2,3], Fátima Cuadrado[1,2,3]

**1** Department of Psychology, University of Cordoba, Cordoba, Andalusia, Spain, **2** Maimonides Biomedical Research Institute of Cordoba (IMIBIC), Cordoba, Andalusia, Spain, **3** Reina Sofía University Hospital of Cordoba, Cordoba, Andalusia, Spain, **4** Early Childhood Care Centre, University of Cordoba, Cordoba, Andalusia, Spain

☯ These authors contributed equally to this work.

* l72varuj@uco.es

**Data Availability Statement:** All data files are available from the OSF database (URL: https://mfr. osf.io/render?url=https%3A%2F%2Fosf.io% 2Fm5xhb%2Fdownload).

## Abstract

Autism Spectrum Disorders (ASD) are associated to social attention (SA) impairments. A gaze bias to non-social objects over faces has been proposed as an early marker of ASD. This bias may be related to the concomitant circumscribed interests (CI), which question the role of competing objects in this atypical visual behavior. The aim of this study was to compare visual attention patterns to social and non-social images in young children with ASD and matched typical controls (N = 36; age range 41–73 months) assessing the role of emotion in facial stimuli and the type of competing object. A paired preference task was designed pairing happy, angry, and neutral faces with two types of objects (related or not related to autism CI). Eye tracking data were collected, and three indexes were considered as dependent variables: prioritization (attentional orientation), preference, and duration (sustained attention). Results showed that both groups had similar visual pattern to faces (prioritization, more attention and longer visits to faces paired with objects non-related to their CI); however, the ASD group attended to faces significantly less than controls. Children with ASD showed an emotional bias (late orientation to angry faces and typical preference for happy faces). Finally, objects related to their CI captured attention in both groups, significantly reducing SA in children with ASD. Atypical SA is present in young children with ASD regardless the competing non-social object. Identifying strengths and difficulties in SA in this population may have substantial repercussion for early diagnosis, intervention, and ultimately prognosis.

## Introduction

Autism Spectrum Disorders (ASD) comprise a set of neurodevelopmental and pervasive conditions in which social-emotional and communication impairments along with restricted and repetitive behaviors are core symptoms [1–4]. The wide heterogeneity of the autism spectrum hampers early diagnosis, which is considered an important predictor of future outcomes and

**Funding:** The authors received no specific funding for this work.

**Competing interests:** The authors have declared that no competing interests exist.

prognosis as it leads to the early intervention and the deployment of appropriate supports [2, 5–10]. In this sense, research is claiming for the study of new markers to improve the early identification of this disorder. The study of social attention (SA, the allocation of visual attention to social stimuli or socially relevant content of scenes instead of non-social elements) has received increasing attention of research on ASD in the last decades. Likewise, new approaches and methodologies such as eye tracking have become widespread due to their significant contribution to the developmental understanding of SA.

Eye tracking methodology allows to analyse individuals' gaze behavior by tracing and monitoring their eye movements during image visualisation. This methodology draws on corneal reflection technology, so it yields reliable attentional markers in a non-invasive way [11–13]. This asset alongside its high sensitivity to detect subtle biases in visual emotion processing and disentangle the core mechanisms of facial emotional expression decoding have increased the use of eye tracking in research on ASD [14–17].

Most studies on SA in children with ASD with eye tracking have reported reduced attention to social images or the social content of visual scenes in favour of an attentional bias to non-social stimuli [15, 18–24]. This atypical attention to social events has been found in toddlers with ASD from the first year of life and is usually accompanied by difficulties in attention disengagement [15, 25] and lack of arousal modulation towards others' emotions [24]. Moreover, visual preference for geometric patterns instead of social images has been suggested as a potential marker for early diagnosis of ASD [26–29]. The scanning pattern (also known as gaze or looking pattern, the sequence of looking shifts during images visualization) has also been studied in children with ASD. Findings in this area yielded a dispersed scanning pattern among individuals with ASD when viewing social images, that is, they displayed scattered eye movements across the scene when it involved a social situation [30]. In the study on SA, face deserves special attention as it is an important gateway to perceive, recognize, and understand others' inner emotional states [31, 32]. Likewise, attention to faces at early ages is a natural mechanism which leads to brain specialization and, consequently, to social expertise [32–35]. Children with ASD have showed reduced attention to faces [24, 36–38]. However, increased attentional orientation to emotional faces rather than to neutral ones was found in this population (although to a lesser extent than the controls). This may suggest that children with ASD, similar to their typically developing (TD) peers, have emotional sensitivity [37, 39]. Moreover, some studies have reported a bias toward positive emotions and better recognition rates of these emotions in children with ASD, as well as in TD children [40–42]. Familiarity has also been yielded as a potential trigger of pupil reactivity and visual attention to faces in children with ASD [39]. These results suggest that attention to faces is not completely impaired in children with ASD, as it can be modulating considering some variables. Regarding the attention to the core facial features, specific results related to the eyes and the mouth are still inconsistent when considering studies independently; however, sound current reviews agree that children with ASD show atypical face processing, which means that they scan faces differently than TD children [16, 43, 44]. This atypicality has been found in toddlers with ASD as young as two years [45], which have led some researchers to suggest that these children may have reduced their social learning input from early ages, which impact on their brain specialization and, consequently, on their social development at later ages [25, 32, 35, 43–46].

## The role of competing objects in SA

The saliency of competing non-social objects plays a crucial role when assessing SA in children with ASD [25, 46, 47]. It has been suggested that the restricted and recurring bias to non-social elements reported in this population may be related to the same cognitive mechanisms involved

in the repetitive behaviors associated to this disorder, mainly the circumscribed interests (CI, [46–48]). Despite the relevance of the topic, few studies have addressed the role of competing objects and their interaction with social stimuli in SA in children with ASD. A restricted looking behaviour toward CI-related objects (CIO, not to non-related ones or to social stimuli) has been found in young children [46] and in school-aged children with ASD [25]. This behavior was described as circumscribed, perseverative, and detail oriented in both studies.

Additionally, Sasson and Touchstone assessed SA in young children with ASD (age range 24–62 months) who performed a paired preference task where faces with different emotions (happiness, sadness, fear, anger, neutral) were paired with CIO and non-CI-related objects (non-CIO) [47]. No difference between emotions was found in this study but their findings revealed that children with ASD showed the same attentional bias towards faces as their TD peers when the competing non-social objects were unrelated to their CI (non-CIO), thus only CIOs were able to capture their attention more than faces.

Given that the presence of CI is a hallmark of ASD and that this restricted looking behaviour seems to be also specific of the autistic phenotype, the effect of the saliency of competing non-social objects on SA may help identify children with ASD at earlier ages.

## Aims and hypotheses

Taken together, previous results showed the high potential of studies on SA in this population as they have yielded some specific markers of the autism spectrum. Thus, the particular role of CIO in SA in children with ASD may also contribute to early diagnosis. Following this rationale and the approach applied in [47], the aim of this study was to compare young children with ASD and their TD peers in terms of SA assessing the role of facial emotions and competing objects in both groups. As in [47], we designed an eye tracking paired preference task where a social stimulus (happy, angry, or neutral face) and a non-social one (CIO or non-CIO) competed for attracting participants' SA. This experiment was based on the approach applied in [47] in terms of the application of the paired preference paradigm and the eye tracking dependent variables definition; however, both studies differs in aspects of design such as the number of emotions considered ([47] comprised happiness, sadness, anger, fear, and neutral, while this study included happiness, anger, and neutral), the manipulation of emotion intensity (in [47] researchers manipulated that variable, while in this experiment intensity was not considered), and the number of trials per condition ([47] included 1trial, while we repeated each condition 3 times). Based on previous studies, we hypothesized that young children with ASD would show significantly reduced attention to faces compared with objects, which would differentiate them from the TD group (H1). We included faces with different emotional expressions (happiness, anger, and neutral) to see the effect of emotionality in SA. Thus, we expected that both groups would pay more attention to emotional faces, particularly to happy faces (H2), due to the reported emotional sensitivity [37, 39] and bias toward positive emotions in children with ASD [40–42]. This would imply that positive emotions had an attracting effect which could be relevant to consider for clinical practice. We also predicted that young children with ASD would pay atypically less attention to faces when they competed with CIO, but typical attention when they competed with non-CIO (H3). Finally, we expected to find a significant bias toward CIO with respect to non-CIO in both groups (H4).

## Method

### Participants

Thirty-eight pre-schoolers: 19 with ASD (18 boys, 1 girl; $M_{age}$ = 55.89 months, $SD_{age}$ = 9.40, range = 44–72 months) and 19 TD (18 boys, 1 girl; $M_{age}$ = 53.53 months, $SD_{age}$ = 9.28,

range = 41–72 months) participated in this study. This sample size could detect between-groups differences with 85% of power and an effect size of 0.5, according to a sensitivity analysis in GPower 3.1.9.7 [49, 50]. The clinical group was recruited from centres of early childhood intervention in the province of Córdoba. For the clinical group, inclusion criteria comprised 1) the attendance to a centre of early childhood intervention, 2) having received a thorough assessment by a licensed experienced clinician who had determined the presence of ASD according to the DSM-5 criteria and following the protocol of Infant Mental Health program at a community mental health service, and 3) the absence of any mental or medical condition.

Children of the TD group were recruited from the preschool classes at a public school in Córdoba. Inclusion criteria for this group encompassed not having a history of developmental disorder either now or in the past, as well as the gender and chronological age matching with the clinical group. As in [47], we matched groups on chronological age instead of on developmental age due to the low demand of the task applied, as passive-observational tasks do not require high-level cognitive abilities. We also matched groups on gender to control its potential effect on CI [47, 51–53].

## Stimuli

Following the approach of [47], the eye tracking paired preference task consisted of pairing social and non-social images to assess visual attention patterns for faces and objects. We paired faces displaying three different emotions (happiness, anger, and neutral) with two types of objects (CIO and non-CIO), which yielded a total of six experimental conditions which were repeated six times using different facial identities (a total of 36 trials). The gender of the faces and the location in the screen were counterbalanced to avoid potential effects of both variables. Thus, 36 pictures from 12 different identities (the same identities were taken for the three emotions, but these were not repeated for the same emotion) were taken from the Amsterdam Dynamic Facial Expression Set (ADFES; [54]). Faces were paired with 36 images of objects (18 CIO and 18 non-CIO, see Fig 1A and 1B). Some of these images were of our own creation by taking pictures of ordinary toys and puzzles used in the Early Childhood Intervention Centre associated to the University of Córdoba, while others were taken from the Pixabay website and

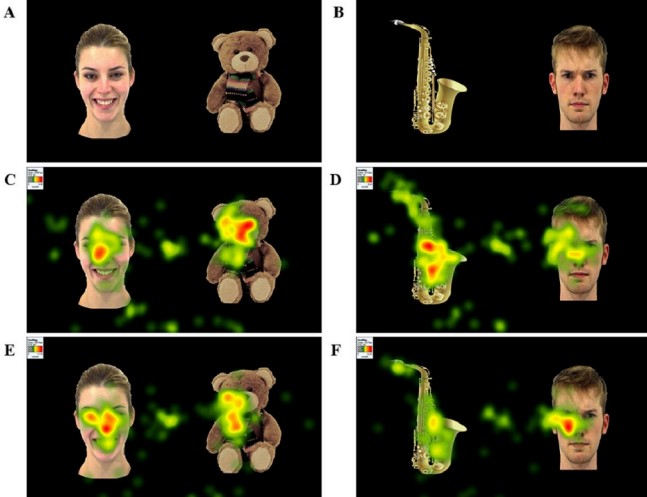

**Fig 1. Examples of stimuli and heatmaps in ASD and TD groups.** (A) Example stimulus pairing a happy face and a CIO. (B) Example stimulus pairing a non-CIO and an angry face. (C and D). Heatmaps of the ASD group performance. (E and F) Heatmaps of the TD group performance.

free of copyright under the Creative Commons CC0 license. Inclusion criteria for CIO were based on previous studies on CI-related topics and objects in ASD [25, 47, 55]. Thus, CIO belonged to the categories of toys, puzzles, means of transport, animals, and blocks, while non-CIO were plants, furniture, musical instruments, tools, school material, and clothes.

Pairs were designed using power point to readjust images size (to 15 x 15 cm approximately), control the screen location, and remove pictures background to avoid contrast with the black background. The background removal forced us to rescale all pictures so that pairs looked similar in size. Final slides were resized to 1500 x 843 pixels to be correctly displayed in the laptop used to assess children and presentation order was counterbalanced.

## Procedure

This study had the approval of the Research Ethics Committee of Córdoba. We contacted with potential candidate centres of early childhood intervention and those interested in participating distributed the project information and the written informed consent between families. Once we collected signed written informed consent from parents or guardians of children, we scheduled the assessment sessions according to families' and centres' availability. The assessment sessions took place in children's natural environment using a quiet room without distractions. In most cases, children were alone with the researcher during the assessment; however, there were some cases in which any parent was present, either for child requirement or for their own request. Assessment sessions were carried out in one single session in all cases, although the option of dividing the evaluation in two sessions had also been offered. The entire process lasted 45 minutes approximately. First, children performed the paired preference task, which lasted 3.6 minutes. Eye tracking data were recorded using Tobii X2-30 (Tobii Technology AB, Stockholm, Sweden) which tracks eye movements at a sampling rate of 30 Hz with a spatial accuracy of 36°. Children were seated at a deemed distance of 60 cm from a 15'screen of a laptop. They were given no other instruction but to look at the screen. The task started with a nine-points calibration with an animated stimulus as target. Calibration process was repeated in those cases where a child failed to complete it. After calibration, the task consisted of visualizing the set of 36 slides showing one face and one object. Each slide was displayed for 5 s. Prior to the presentation of each slide, children viewed an animated fixation point (a cartoon) for 1 s. to drive their attention to the centre of the screen. After the eye tracking task, we assessed receptive vocabulary, affect recognition, and Theory of Mind (ToM) as described in the next section.

## Measures

Along with the eye tracking task, receptive vocabulary, affect recognition, and Theory of Mind (ToM) abilities were assessed. using the Peabody Picture Vocabulary Test-Third Edition (PPTV-III, [56]) and the two subtests comprised in the social perception domain of the Developmental Neuropsychological Assessment (NEPSY-II, [57]) to analyse potential correlations among these domains and visual attention patterns in the sample. These scales were chosen due to their wide usability for clinical and research purposes, and the fact that both yield typification data with special groups (including ASD).

PPTV-III is a screening test which assesses vocabulary comprehension in individuals from 2.6 to 90 years. Subjects are presented with an array of four pictures and are asked to choose which one fits to a given word. Individuals are allowed to point the finger at their answer to reduce expressive language demands.

NEPSY-II is a neuropsychological assessment battery which comprises 32 tests to assess six domains (attention and executive functioning, language, memory and learning, sensorimotor,

social perception, and visuospatial processing) in children and adolescents from 3 to 16 years. In the present study, only social perception domain was assessed. This domain encompasses affect recognition and ToM. Some participants of the ASD group were not able to perform either any of the two tasks or only the ToM part because of their age or the high linguistic demand required to pass these tasks.

Following the approach of [47], we defined three indexed as dependent variables: 1) Prioritization, time to first fixation to faces to assess attentional orientation, 2) Preference, the proportion of total fixation duration on each facial AOI to compare the distribution of visual attention between faces regarding the competing object, and 3) Duration, mean time per visit to faces to describe the gaze maintenance (sustained attention) in facial stimuli. Statistical tests were performed using the Statistical Package for the Social Sciences, 25 (SPSS, 25, IBM, Armonk, NY, United States of America).

## Data analysis

To analyse visual attention pattern to faces in both groups, we conducted separate repeated measures ANOVAs on each dependent variable (Prioritization, Preference, and Duration), with Group (ASD, control) as the between-group factor and Type of Object (CIO, non-CIO) as the within-group factor. Likewise, to examine visual attention pattern to objects in each group, we again performed separate repeated measures ANOVAs on each dependent variable (Prioritization, Preference, and Duration), with Group (ASD, control) as the between-group factor and Type of Object (CIO, non-CIO) as the within-group factor. As groups significantly differed in receptive vocabulary (see Table 1), PPVT-II standard scores were tentatively included as covariates, but preliminary analyses yielded no effect, therefore this measure was dropped from the final analyses. Effect sizes were also calculated for repeated measures ANOVAs (partial eta-squared, $\eta_p^2$) considering small ($< .01$), medium ($< .06$), or large ($< .14$) effects. Finally, correlation analyses were carried out to check potential relationships between visual attention pattern and PPVT-III or NEPSY-II scores in any group. Thus, Pearson correlation coefficient was used for PPVT-III scores, while Spearman correlation coefficient was used for NEPSY-II scores as this measure did not fulfil requirements for performing parametric tests.

## Results

### Descriptive characteristics of variables of the study

All participants assessed were included in the statistical analysis as the eye tracker apparatus collected more than 50% of their fixations (which is a standard cut-point applied in many studies [26, 28, 29, 36, 58]), thus all the 38 children were suitable for the analysis. Sample characteristics are summarized in Table 1. No significant differences were found between groups in age or gender (p > .05). However, ASD and TD groups differed in terms of vocabulary comprehension (PPVT-II standard scores), emotion recognition abilities (NEPSY-II Total Score Affect Recognition) and ToM (Nepsy-II Verbal ToM and Total Score ToM).

### Social attention

**Prioritization.** No effect for group × type of object interaction was found in any emotion. A main effect for group was only found in angry faces ($F_{(1, 37)} = 4.54$, $p < .04$, $\eta_p^2 = .11$), indicating that children with ASD looked at angry faces significantly later than TD children (see Fig 2B). A main effect for type of object was significant in all emotions independently: happiness ($F_{(1, 37)} = 5.17$, $p < .03$, $\eta_p^2 = .13$), anger ($F_{(1, 37)} = 10.45$, $p < .00$, $\eta_p^2 = .23$), neutral ($F_{(1,}$

**Table 1. Descriptive characteristics of variables of the study.**

|  |  |  | ASD (n = 19) | TD (n = 19) |  |  |
|---|---|---|---|---|---|---|
|  |  |  | *M (SD)* | *M (SD)* | $F_{(1, 37)}$ | *P* |
| PPVT-III Raw Score |  |  | 43.05 (19.54) | 50.21 (18.73) | 1.33 | .257 |
| PPVT-III Standard Score |  |  | 94.00 (25.85) | 107.47 (11.38) | 4.33 | .045 |
|  |  |  | n | n | $\chi^2$ | *P* |
| Nepsy-II Total Raw Score Affect Recognition |  |  | 7 | 19 | 4.03 | .045 |
| Nepsy-II Verbal ToM |  |  | 6 | 19 | 7.55 | .006 |
| Nepsy-II Contextual ToM |  |  | 6 | 19 | 1.15 | .283 |
| Nepsy-II Total Raw Score ToM |  |  | 6 | 19 | 8.27 | .004 |
|  |  |  | ASD (n = 19) | | TD (n = 19) | |
|  |  |  | *M (SD)* | | *M (SD)* | |
| Prioritization (ms) | Faces | TFF_Hapvs.CIO | 1198 (385) | | 1077 (421) | |
|  |  | TFF_Hapvs.non-CIO | 957 (464) | | 878 (403) | |
|  |  | TFF_Angvs.CIO | 1287 (420) | | 969 (395) | |
|  |  | TFF_Angvs.non-CIO | 956 (392) | | 814 (415) | |
|  |  | TFF_Neuvs.CIO | 1231 (544) | | 1076 (551) | |
|  |  | TFF_Neuvs.non-CIO | 950 (452) | | 795 (417) | |
|  |  | TFF_Facvs.CIO | 1239 (321) | | 1041 (362) | |
|  |  | TFF_Facvs.non-CIO | 954 (290) | | 827 (262) | |
|  | Objects | TFF_CIOvs.Happ | 933 (551) | | 937 (365) | |
|  |  | TFF_non-CIOvs.Hap | 735 (351) | | 961 (344) | |
|  |  | TFF_CIOvs.Ang | 752 (427) | | 758 (312) | |
|  |  | TFF_non-CIOvs.Ang | 830 (370) | | 1079 (392) | |
|  |  | TFF_CIOvs.Neu | 779 (425) | | 754 (342) | |
|  |  | TFF_nonCIOvs.Neu | 696 (395) | | 1281 (388) | |
|  |  | TFF_TotalCIO | 820 (369) | | 816 (215) | |
|  |  | TFF_Totalnon-CIO | 753 (207) | | 1107 (236) | |
| Preference (%) | Faces | PTFD_Hapvs.CIO | 14 (11) | | 17 (4) | |
|  |  | PTFD_Hapvs.non-CIO | 16 (9) | | 19 (6) | |
|  |  | PTFD_Angvs.CIO | 14 (7) | | 17 (6) | |
|  |  | PTFD_Angvs.non-CIO | 17 (9) | | 22 (5) | |
|  |  | PTFD_Neuvs.CIO | 10 (5) | | 16 (3) | |
|  |  | PTFD_Neuvs.non-CIO | 14 (4) | | 19 (6) | |
|  |  | PTFD_Facvs.CIO | 38 (13) | | 49 (10) | |
|  |  | PTFD_Facvs.non-CIO | 47 (15) | | 59 (11) | |
|  | Objects | PTFD_CIOvs.Happ | 22 (6) | | 18 (4) | |
|  |  | PTFD_non-CIOvs.Hap | 17 (7) | | 13 (5) | |
|  |  | PTFD_CIOvs.Ang | 19 (7) | | 15 (4) | |
|  |  | PTFD_non-CIOvs.Ang | 15 (6) | | 14 (6) | |
|  |  | PTFD_CIOvs.Neu | 21 (7) | | 18 (5) | |
|  |  | PTFD_nonCIOvs.Neu | 21 (7) | | 14 (6) | |
|  |  | PTFD_TotalCIO | 62 (13) | | 51 (10) | |
|  |  | PTFD_Totalnon-CIO | 53 (15) | | 41 (11) | |
| Duration (ms) | Faces | TPV_Hapvs.CIO | 638 (214) | | 992 (317) | |
|  |  | TPV_Hapvs.non-CIO | 651 (315) | | 980 (405) | |
|  |  | TPV_Angvs.CIO | 646 (323) | | 986 (384) | |
|  |  | TPV_Angvs.non-CIO | 728 (292) | | 1106 (322) | |
|  |  | TPV_Neuvs.CIO | 545 (208) | | 802 (197) | |
|  |  | TPV_Neuvs.non-CIO | 669 (453) | | 958 (340) | |
|  |  | TPV_Facvs.CIO | 615 (171) | | 912 (235) | |
|  |  | TPV_Facvs.non-CIO | 655 (201) | | 1016 (274) | |

*(Continued)*

**Table 1.** (Continued)

| | | | | |
|---|---|---|---|---|
| **Objects** | **TPV_CIOvs.Happ** | 1203 (768) | 965 (336) |
| | **TPV_non-CIOvs.Hap** | 1222 (1504) | 697 (313) |
| | **TPV_CIOvs.Ang** | 1480 (2779) | 745 (250) |
| | **TPV_non-CIOvs.Ang** | 1578 (3382) | 792 (344) |
| | **TPV_CIOvs.Neu** | 1335 (1233) | 927 (283) |
| | **TPV_nonCIOvs.Neu** | 1144 (1145) | 816 (295) |
| | **TPV_TotalCIO** | 1157 (896) | 858 (211) |
| | **TPV_Totalnon-CIO** | 1128 (1426) | 761 (225) |

PPVT-III Raw Score and Standard Score variables were tested using ANOVA. Nepsy-II scores were analysed using H-Kruskal-Wallis.

TFF, Time to First Fixation; PTFD, Proportion of Total Fixation Duration; TPV, Time per Visit.

$_{37)}$ = 6.47, $p < .02$, $\eta_p^2 = .15$), and in total faces ($F_{(1, 37)} = 22.38$, $p < .00$, $\eta_p^2 = .38$), implying that all participants looked at faces paired with non-CIOs earlier than those paired with CIOs (see Figs 2A and 2B and **3A**).

**Preference.** No group × type of object interaction effect was yielded in any particular emotion or total faces, but a main effect for group was found in angry ($F_{(1, 37)} = 7.40$, $p < .01$, $\eta_p^2 = .17$), neutral ($F_{(1, 37)} = 14.78$, $p < .00$, $\eta_p^2 = .29$), and total faces ($F_{(1, 37)} = 10.29$, $p < .00$, $\eta_p^2 = .22$), indicating that children with ASD attended to angry and neutral faces substantially less than their TD peers (see Figs 1D, 1F, 2D and 3C). A main effect for type of object also emerged in angry ($F_{(1, 37)} = 5.97$, $p < .02$, $\eta_p^2 = .14$), neutral ($F_{(1, 37)} = 16.83$, $p < .00$, $\eta_p^2 = .32$), and total faces ($F_{(1, 37)} = 31.57$, $p < .00$, $\eta_p^2 = .47$), implying that both groups paid significantly more attention to angry and neutral faces paired with non-CIOs than those paired with CIOs. Any effect was found in happy faces, which suggests that both groups looked at these faces the same amount of time regardless the type of object (see Figs 1C, 1E and 2C).

**Duration.** No group × type of object interaction effect was highlighted in any specific emotion or total faces, but a main effect for group was found in all emotions: happy ($F_{(1, 37)} = 13.66$, $p < .00$, $\eta_p^2 = .28$), angry ($F_{(1, 37)} = 14.41$, $p < .00$, $\eta_p^2 = .29$), neutral ($F_{(1, 37)} = 11.80$, $p < .00$, $\eta_p^2 = .25$), and total faces ($F_{(1, 37)} = 24.08$, $p < .00$, $\eta_p^2 = .40$), indicating that children with ASD made significantly shorter visits to all type of faces than their TD peers. A main effect for type of object was also found in neutral ($F_{(1, 37)} = 4.62$, $p < .04$, $\eta_p^2 = .11$), and total faces ($F_{(1,}$

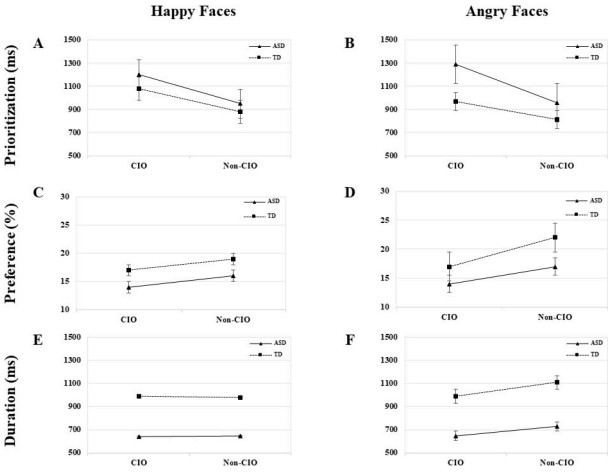

**Fig 2. Visual attention patterns in happy and angry faces.**

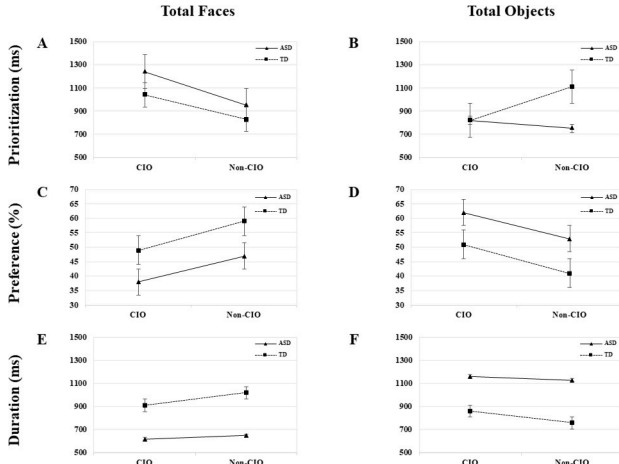

**Fig 3. Visual attention patterns in total faces and objects.**

$_{37)} = 6.59$, $p < .02$, $\eta_p^2 = .16$). This implies that the lack of facial emotion drove both groups to pay more attention to faces paired with non-CIOs than those paired with CIOs; otherwise, when faces displayed any positive or negative emotion, the type of object had no impact in their sustained attention (see Figs 2E, 2F and 3E).

## Object attention

**Prioritization.** Significant effects were found for group × type of object interaction ($F_{(1, 37)} = 12.85$, $p < .00$, $\eta_p^2 = .26$), group ($F_{(1, 37)} = 6.26$, $p < .02$, $\eta_p^2 = .15$), and type of object ($F_{(1, 37)} = 5.08$, $p < .03$, $\eta_p^2 = .12$), indicating that, relative to TD, children with ASD were significantly faster at focusing on non-CIOs (see Fig 3B). These effects were replicated only when objects were paired with neutral faces, but not when these were paired with happy or angry faces, implying that both groups showed similar orientation to objects when these competed with emotional faces.

**Preference.** No group × type of object interaction was found in this index, but a main effect emerged for group ($F_{(1, 37)} = 10.29$, $p < .00$, $\eta_p^2 = .22$), indicating that children with ASD looked at objects substantially longer than their counterparts (see Fig 3D). An effect for type of object was also found ($F_{(1, 37)} = 31.57$, $p < .00$, $\eta_p^2 = .47$), implying that both groups spent more time viewing CIOs than non-CIOs. When considering emotions, these effects were replicated only for happy and neutral faces, but these were marginal for angry faces, which suggests that children with ASD showed typical preference for objects when they competed with angry faces.

**Duration.** No significant effect was found at any level in this index, indicating that children with ASD as well as TD made similar visits to objects, regardless its typology (see Fig 3F).

## Correlation analysis

To explore whether the visual attention pattern found in each group was related to their performance in the PPVT-III and the NEPSY-II, we conducted correlation analyses. No significant correlation between PPVT-III and any dependent variable was found in any group. Only Spearman correlation coefficient yielded significant values (see Table 2) for Affect Recognition Total Raw Score and prioritization of happy faces ($r_{s(7)} = .775$, $p = .04$) in the ASD group. For the TD group, a significant correlation was found in Affect Recognition Total Raw Score and

**Table 2. Spearman correlation between dependent variables and NEPSY-II indexes.**

| | | ASD (n = 7) | | | | TD (n = 19) | | | |
|---|---|---|---|---|---|---|---|---|---|
| | | TRSAF | VT | CT | TRST | TRSAF | VT | CT | TRST |
| **Prioritization** | TFF Happy Faces | .775* | .309 | .463 | .486 | -.251 | -.350 | -.211 | -.376 |
| | TFF Angry Faces | .054 | .093 | -.802 | -.200 | -.027 | -.030 | -.129 | -.070 |
| | TFF Neutral Faces | .355 | .313 | -.204 | .232 | .171 | -.131 | -.129 | -.154 |
| | TFF CIO | .487 | .278 | .617 | .486 | .588** | .329 | .303 | .341 |
| | TFF nonCIO | .414 | .278 | .309 | .371 | .218 | .124 | .190 | .371 |
| **Preference** | PTFD Happy Faces | -.545 | -.309 | .000 | -.314 | -.145 | .100 | .351 | .226 |
| | PTFD Angry Faces | -.255 | -.517 | -.125 | -.493 | .023 | .086 | .056 | .054 |
| | PTFD Neutral Faces | -.093 | -.751 | .031 | -.638 | .238 | -.049 | .424 | .086 |
| | PTFD CIO | .055 | .494 | .185 | .486 | -.092 | -.151 | -.295 | -.191 |
| | PTFD nonCIO | .577 | .525 | .309 | .600 | .000 | -.053 | -.267 | -.153 |
| **Duration** | TPV Happy Faces | .036 | -.278 | .278 | -.143 | .044 | .264 | .200 | .344 |
| | TPV Angry Faces | .270 | .062 | .031 | .086 | .287 | .345 | .054 | .315 |
| | TPV Neutral Faces | -.288 | -.617 | -.463 | -.714 | .379 | .124 | .207 | .146 |
| | TPV CIO | .306 | .525 | .000 | .486 | .105 | -.115 | .176 | -.011 |
| | TPV nonCIO | .360 | .741 | .309 | .771 | .178 | .085 | -.035 | .080 |

TRSAF, Total Raw Score Affect Recognition; VT, Verbal ToM; CT, Contextual ToM; TRST, Total Raw Score ToM; TFF, Time to first fixation; PTFD, Proportion of total fixation duration; TPV, Time per visit. Significance levels: ***$p < 0.001$; **$p < 0.01$; and *$p < 0.05$.

TFF to CIO ($r_{s(19)}$ = .588, $p$ = .01). This data indicates that: 1) the higher Affect Recognition scores children with ASD have, the later they look at happy faces; 2) the higher Affect Recognition scores TD children have, the later they focus on CIO.

## Discussion

This study aimed at comparing SA in young children with ASD and TD peers analysing the effect of competing facial emotions and type of objects in their visual attention pattern to faces and objects. Taken together, our results suggest that children with ASD display a visual attention pattern typical in terms of direction (early orientation, longer looking time, and more detailed exploration of faces paired with non-CIOs than those paired with CIOs, and visual preference for CIOs over non-CIOs), but quantitatively atypical. Thus, relative to TD children, children with ASD showed a lack of attention to faces and an excessive visual preference for objects in general.

Our results revealed that children with ASD looked at faces later, during less time and they made shorter visits than their TD peers. In light of these results, our first hypothesis was confirmed as children with ASD differed from TD children in reduced attention to faces. This statement aligns with many studies reporting reduced attention to social images in favour of non-social ones [15, 18–24] and, particularly, decreased attention to faces [24, 36–38].

We also predicted different effects of competing facial emotions in children's visual behavior (H2). These differences were found in prioritization and preference indexes during the analysis of the visual attention pattern to faces, confirming emotional sensitivity in children with ASD. Thus, our ASD group looked at angry faces later than the control one, indicating late orientation to angry faces compared to controls. Moreover, happy faces were the only ones in which children with ASD displayed a typical visual preference behavior, which implies that happiness was relatively unimpaired in our ASD sample. Likewise, the effect of the type of

object in sustained attention to faces in children with ASD was only significant when objects competed with neutral faces, which suggests that the absence of emotion increases atypical visual behavior in this population. This emotional sensitivity was again replicated in the analysis of visual attention pattern to objects, as children with ASD only showed atypical visual orientation to objects when they competed with neutral faces, otherwise there were no differences between groups, type of object, or interaction. These assumptions are in line with those studies reporting emotional sensitivity [37, 39] and highlighting that negative emotions are compromised in this population, while positive emotions are relatively intact [40–42, 59, 60]. This finding is worthy to be considered when designing and developing interventions. Hence, clinicians may avoid negative emotions (which drive to attention disengagement) and foster positive ones (which are attention-getters); this may help involve the child within the activity, increasing the effectiveness and potential benefits of interventions. Emotional sensitivity was not reported in [47], this difference between studies could be due to the fact that we included more essays per condition, which may have contributed to uncover atypicalities in visual behavior to different emotional faces in the ASD group.

Regarding our H3, we expected to find a differential role of the type of object in young children with ASD, which would drive them to display less attention to faces competing with CIO, but typical attention when they competed with non-CIO (H3). This hypothesis was refused in this study as children with ASD paid less attention to faces in both conditions. Moreover, TD children seemed to identify quickly the type of object they were looking at, as they focused on CIOs significantly earlier than on non-CIOs; however, children with ASD did not make this distinction, instead they fixated on both type of objects equally fast. Therefore, the type of object was not what made the greatest impact on SA in children with ASD, but the competing facial emotion was. This disagreement with conclusions from [47] may rely on the age of participants in both studies (being ours older) or the fact that our participants had been receiving early intervention for a deemed mean time of 30 months ($SD$ = 8 months), which implies a previous work on their potential social-emotional deficits. In this sense, future studies should address the effect of early diagnosis and intervention in SA in children with ASD.

We also found a powerful effect of CIOs over non-CIOs in both groups, this effect was found in the three indexes when analysing visual attention pattern to faces, but only in prioritization and preference when considering the visual attention pattern to objects. The lack of differences in duration index during the analysis of visual attention pattern to objects could be partially explained by the fact that the ASD group was more heterogeneous in terms of duration and number of visits to both type of objects. This shows the wide heterogeneity of CI among children with ASD due to their idiosyncratic feature. Due to the close relationship between the duration index and the visual processing style, we may say that children with ASD made less detailed processing of faces compared to their TD peers, but not a different visual processing of objects. Still, these data confirm our H4 suggesting that CIOs disrupt social attention in children with ASD as well as TD. This result is in line with a large body of literature on the catching role of CIOs children with ASD and TD [25, 46, 47, 51–53, 55]. Nevertheless, given the fact that children with ASD have difficulties engaging with faces, we think that clinicians should be cautious including CIOs in interventions as they can help engage the child in an activity, but they can also distract him/her from its real purpose.

On the other hand, no significant correlation between vocabulary comprehension or ToM and visual attention pattern was highlighted in any group. Only affect recognition was significantly related to prioritization of happy faces and CIO in the ASD group and TD group, respectively. This indicates that the more proficiency in affect recognition, the later children

with ASD look at happy faces and TD children look at CIO. This outcome is reasonable in the case of TD children as affect recognition is closely related to social attention [31, 32, 61], thus more proficiency in affect recognition could imply diminishing attention to non-social stimuli in favour of social ones. However, the pattern found in ASD is surprising and cannot be explained based on existing literature; hence it could stem from the small sample of children with ASD to whom NEPSY-II could be applied.

Finally, we consider some limitations of this study. First of all, the sample size was relatively small, although it is in line with that of related studies [22, 30, 37, 38, 45, 47]; studies with larger samples may help to stress differences between groups and remark patterns within-group. Secondly, we controlled for receptive language, affect recognition, and ToM abilities, but not for expressive language, which would have been interesting given its key role in this population [62]. Moreover, the small number of children with ASD who could not be assessed with the social-emotions subscale of the NEPSY-II may have distorted results on correlations between variables, as some studies have highlighted the close relationship between affect recognition and SA [31, 32, 61] as well as facial emotion recognition and ToM [63, 64] in this population. Alongside these limitations, it is important to be cautious when interpreting results of SA in children with ASD as some variables may impact on them. Among these variables, research has highlighted the following: 1) the stimulus type, being interactive and naturalistic stimuli those which better find differences between clinical and non-clinical groups [18, 65]; 2) the social content of scenes, that is, the more people involved in the scene or the more scene complexity, the less SA deployed by the ASD population [20, 21, 58]; 3) the presence of speech during social interactions, which has been related to slower latencies and less social orientation to faces in children with ASD compared to TD [66]; and 4) the language proficiency, with typical SA in children with ASD and normal language but reduced SA in those with language delayed [62]. For these reasons, futures studies must include larger sample sizes, a more thorough assessment of language profile of the clinical sample, as well as affect recognition and ToM measures, and more comparison groups to contribute to differential diagnosis.

## Conclusions

Children with ASD have demonstrated SA atypicalities which may impair their relationship with the social environment and, consequently, their social expertise from early ages. Current eye tracking research on SA has been aimed at identifying these atypicalities in young children, which has substantial repercussion in clinical practice. In this sense, the finding of specific visual attentional markers such as the less detailed processing of faces and the bias to non-social objects in the ASD population are valuable from a diagnosis point of view, but also the emotional sensitivity found in these children is significant from an intervention perspective, as it may help practitioners to guide their interventions emphasizing the role of emotions, taking advantage of those more engaging for children with ASD, and avoiding the most repelling ones. The worthy applicability of this kind of studies alongside the widely reported advantages of early intervention for children future outcomes and development substantiate the need of continuing this line of research enriching it with larger samples, deeper cognitive assessments, and more comparison groups.

## Acknowledgments

We gratefully acknowledge to all families who agreed that their children took part in this study, as well as the Early Intervention Centres and the school which also accepted our request of participation very kindly.

## Author Contributions

**Conceptualization:** Julia Vacas, Adoración Antolí, Araceli Sánchez-Raya, Carolina Pérez--Dueñas, Fátima Cuadrado.

**Data curation:** Julia Vacas.

**Formal analysis:** Julia Vacas, Adoración Antolí, Araceli Sánchez-Raya, Carolina Pérez--Dueñas, Fátima Cuadrado.

**Methodology:** Julia Vacas.

**Resources:** Julia Vacas, Adoración Antolí, Araceli Sánchez-Raya, Carolina Pérez-Dueñas, Fátima Cuadrado.

**Supervision:** Adoración Antolí, Araceli Sánchez-Raya, Carolina Pérez-Dueñas, Fátima Cuadrado.

**Writing – original draft:** Julia Vacas.

**Writing – review & editing:** Julia Vacas, Adoración Antolí, Araceli Sánchez-Raya, Carolina Pérez-Dueñas, Fátima Cuadrado.

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
