## [Decision Letter · Decision Letter 0]

16 Feb 2021

PONE-D-20-36078

Visual  preference for social vs. non social images in young children with autism spectrum disorders. An eye tracking study

PLOS ONE

Dear Dr. Vacas Ruiz,

Thank you for submitting your manuscript to PLOS ONE. After careful consideration, we feel that it has merit but does not fully meet PLOS ONE’s publication criteria as it currently stands. Therefore, we invite you to submit a revised version of the manuscript that addresses the points raised during the review process.

Please, follow carefully recomendations by Reviewer #1. Likewise, I encourage the authors to make the next two amendments as a priority:

   a) In the abstract, where you say "Children with ASD showed an emotional bias (avoidance of angry faces and typical preference for happy faces)", this should be corrected since it does not follow from the results, as longer reaction time is not necessarily an indication of reduced interest or avoidance drive. It may simply mean greater difficulty in processing the more complex information involved in an expression of anger.

  b) In page 16 (lines 346-347), where you say "Thus, our ASD group looked at angry faces later than the control one, indicating fast identification of anger and some extent of avoidance of this emotion", this should be corrected too. Where it says "fast identification" should read "low identification", I guess. And the second part of that sentence (i.e. "and some extend of avoidance of this emotion") should be deleted, since that is a conclusion not derived from data, but speculation.

We look forward to receiving your revised manuscript.

Kind regards,

Juan-Carlos Pérez-González, Ph.D.

Academic Editor

PLOS ONE

Journal Requirements:

2. You indicated that you had ethical approval for your study.

In your Methods section, please ensure you have also stated whether you obtained consent from parents or guardians of the minors included in the study or whether the research ethics committee or IRB specifically waived the need for their consent.

3. Peer review at PLOS ONE is not double-blinded (https://journals.plos.org/plosone/s/editorial-and-peer-review-process). For this reason, authors should include in the revised manuscript all the information removed for blind review, including recruitment locations details.

4. Please change "female” or "male" to "woman” or "man" as appropriate, when used as a noun (see for instance https://apastyle.apa.org/style-grammar-guidelines/bias-free-language/gender).

5. Our internal editors have looked over your manuscript and determined that it is within the scope of our Cognitive Developmental Psychology Call for Papers. The Collection will encompass a diverse range of research articles in developmental psychology, including early cognitive development, language development, atypical development, cognitive processing across the lifespan, among others, with an emphasis on transparent and reproducible reporting practices.  Additional information can be found on our announcement page: https://collections.plos.org/s/cognitive-psychology. 

If you would like your manuscript to be considered for this collection, please let us know in your cover letter and we will ensure that your paper is treated as if you were responding to this call.

Please note that being considered for the Collection does not require an additional peer review beyond the journal’s standard process and will not delay the publication of your manuscript if it is accepted by PLOS ONE.

If you would prefer to remove your manuscript from collection consideration, please specify this in the cover letter.

7. We note that Figure 1 includes images of an individual in the study. 

As per the PLOS ONE policy (http://journals.plos.org/plosone/s/submission-guidelines#loc-human-subjects-research) on papers that include identifying, or potentially identifying, information, the individual(s) or parent(s)/guardian(s) must be informed of the terms of the PLOS open-access (CC-BY) license and provide specific permission for publication of these details under the terms of this license.

Please download the Consent Form for Publication in a PLOS Journal (http://journals.plos.org/plosone/s/file?id=8ce6/plos-consent-form-english.pdf). The signed consent form should not be submitted with the manuscript, but should be securely filed in the individual's case notes.

Please amend the methods section and ethics statement of the manuscript to explicitly state that the patient/participant has provided consent for publication: “The individual in this manuscript has given written informed consent (as outlined in PLOS consent form) to publish these case details”.

Reviewers' comments:

Reviewer's Responses to Questions

**Comments to the Author**

1. Is the manuscript technically sound, and do the data support the conclusions?

Reviewer #1: Yes

2. Has the statistical analysis been performed appropriately and rigorously? 

Reviewer #1: Yes

3. Have the authors made all data underlying the findings in their manuscript fully available?

Reviewer #1: Yes

4. Is the manuscript presented in an intelligible fashion and written in standard English?

Reviewer #1: Yes

5. Review Comments to the Author

Reviewer #1: The social attention impairment is good documented in autism spectrum disorders (ASD). The manuscript deepens in the analysis of this area, comparing visual attention patterns to social and non-social images in young children with ASD and matched typical controls. The study assesses the role of emotion in facial stimuli (happy, angry and neutral faces) and the type of competing object (circumscribed and non-circumscribed interest), using eye tracking.

I consider that the article is consistent with the proposed objectives. It is well written and presents a rigorously methodology. The results finding are relevant, specially the outcome about the emotional sensitivity in children with ASD because their applicability in early interventions.

However, I have a question to authors and propose recommendations to improve the manuscript.

First, the design and presentations of the stimulus follow the approach of Sasson and Touchstone (2014). The variables of study and methodology are the same? Is this article a replication of Sasson and Touhstone research? It’s no clear in the manuscript and I consider necessary inform more detailed about it.

The section Materials and methods could be called only Method. The Materials or Measures are part of Method. The subsections Measures might include the instruments used to evaluate the variables of studies. The dependent variables are included in Data analysis. Changed this at measures section.

In Table 1, the information about age and gender is redundant because this information is in Participants section. It’s better inserts this table in Results in a new section titled Descriptive characteristics of all variables of the study (including dependent variables) in the two groups, eliminating the section participants of the result.

The correlation analysis result it is no clear. I recommend insert a Table with the correlations between variables.

The Discussion with the limitations of the study and References are good.

6. PLOS authors have the option to publish the peer review history of their article (what does this mean?). If published, this will include your full peer review and any attached files.

Reviewer #1: No

---

## [Author Response · Author response to Decision Letter 0]

29 Mar 2021

Response to editor:

1. Editor´s comment. In the abstract, where you say "Children with ASD showed an emotional bias (avoidance of angry faces and typical preference for happy faces)", this should be corrected since it does not follow from the results, as longer reaction time is not necessarily an indication of reduced interest or avoidance drive. It may simply mean greater difficulty in processing the more complex information involved in an expression of anger. 

a. Authors´ response. In the abstract, where we said "Children with ASD showed an emotional bias (avoidance of angry faces and typical preference for happy faces)", we clarified that the emotional bias did not mean avoidance but late orientation to angry faces (Page 2, line 37).

2. Editor´s comment. In page 16 (lines 346-347), where you say "Thus, our ASD group looked at angry faces later than the control one, indicating fast identification of anger and some extent of avoidance of this emotion", this should be corrected too. Where it says "fast identification" should read "low identification", I guess. And the second part of that sentence (i.e. "and some extend of avoidance of this emotion") should be deleted, since that is a conclusion not derived from data, but speculation.

a. Authors´ response. We also corrected the expression ‘fast identification of anger and some extent of avoidance of this emotion’, substituting it by ‘late orientation to angry faces compared to controls’ (Page 19, line 359). 

3. Editor´s comment. Please ensure that your manuscript meets PLOS ONE's style requirements, including those for file naming. 

a. Authors´ response. Following your advice, we have revised PLOS ONE's style requirements and we have made the following style and file naming amendments:

i. We have indented the first line of the first paragraph after headings (page 2, line 24; page 3, line 47; page 5, lines 96 and 117; page 7, lines 145 and 164; page 9, line 192; page 11, line 259; page 14, line 274; page 15, lines 285 and 296; page 16, lines 308 and 316; page 17, lines 325 and 328; page 18, line 342; and page 22, lines 440 and 454).

ii. We have corrected the use of italics in level 3 heading (Page 14, line 273; page 15, lines 284 and 295; page 16, lines 307, 315; and page 17, line 324).

iii. We have corrected figures captions in figures 2 and 3, substituting the word ‘Figure’ by ‘Fig’ (Page 15, lines 282-283).

iv. We have renamed figure files as ‘Fig 1.tif’, ‘Fig 2.tif’, and ‘Fig 3.tif’ instead of ‘Fig 1’, ‘Fig 2’, and ‘Fig 3’.

v. We have capitalized ‘table 1’ in page 11, line 248.

4. Editor´s comment. In your Methods section, please ensure you have also stated whether you obtained consent from parents or guardians of the minors included in the study or whether the research ethics committee or IRB specifically waived the need for their consent.

a. Authors´ response. In the Methods section, we have specified that we obtained signed written consent from parents or guardians of children who took part in the study before carrying out the assessments (Page 9, lines 195-196).

5. Editor´s comment. Peer review at PLOS ONE is not double-blinded (https://journals.plos.org/plosone/s/editorial-and-peer-review-process). For this reason, authors should include in the revised manuscript all the information removed for blind review, including recruitment locations details.

a. Authors´ response. We have added in the revised manuscript all the information removed for blind review, including recruitment locations details (Page 7, lines 150 and 157; page 8, line 175; and page 9, line 192).

6. Editor´s comment. Please change "female” or "male" to "woman” or "man" as appropriate, when used as a noun (see for instance https://apastyle.apa.org/style-grammar-guidelines/bias-free-language/gender).

a. Authors´ response. We have changed the words ‘male’ and ‘female’ by ‘boys’ and ‘girl’ when used as nouns (Page 7, lines 145-146).

7. Editor´s comment. Our internal editors have looked over your manuscript and determined that it is within the scope of our Cognitive Developmental Psychology Call for Papers. The Collection will encompass a diverse range of research articles in developmental psychology, including early cognitive development, language development, atypical development, cognitive processing across the lifespan, among others, with an emphasis on transparent and reproducible reporting practices… If you would like your manuscript to be considered for this collection, please let us know in your cover letter and we will ensure that your paper is treated as if you were responding to this call.

a. Authors´ response. Authors inform that we would be delighted if our manuscript were considered for the Cognitive Developmental Psychology Call for Papers of the journal. 

8. Editor´s comment. We note that you have stated that you will provide repository information for your data at acceptance. Should your manuscript be accepted for publication, we will hold it until you provide the relevant accession numbers or DOIs necessary to access your data. If you wish to make changes to your Data Availability statement, please describe these changes in your cover letter and we will update your Data Availability statement to reflect the information you provide.

a. Authors´ response. We would like to change our Data Availability statement by providing the link to access our data in an open repository: https://mfr.osf.io/render?url=https%3A%2F%2Fosf.io%2Fm5xhb%2Fdownload

9. Editor´s comment. We note that Figure 1 includes images of an individual in the study. As per the PLOS ONE policy (http://journals.plos.org/plosone/s/submission-guidelines#loc-human-subjects-research) on papers that include identifying, or potentially identifying, information, the individual(s) or parent(s)/guardian(s) must be informed of the terms of the PLOS open-access (CC-BY) license and provide specific permission for publication of these details under the terms of this license.

Please download the Consent Form for Publication in a PLOS Journal (http://journals.plos.org/plosone/s/file?id=8ce6/plos-consent-form-english.pdf). The signed consent form should not be submitted with the manuscript but should be securely filed in the individual's case notes.

Please amend the methods section and ethics statement of the manuscript to explicitly state that the patient/participant has provided consent for publication: “The individual in this manuscript has given written informed consent (as outlined in PLOS consent form) to publish these case details”.

a. Authors´ response. We would also like to clarify that Figure 1 includes some examples of the stimuli presented to participants. These stimuli show paired pictures of objects with pictures of faces taken from the Amsterdam Dynamic Facial Expression Set (ADFES), hence Figure 1 does not show any of our participants. The mentioned facial expression set is cited in the method section (stimuli subsection) and its reference appears in the reference section. Examples of pictures from that set are allowed to be included in scientific papers after obtaining written consent from the producers, which we have already collected.

Response to reviewer 1:

1. Reviewer´s comment. First, the design and presentations of the stimulus follow the approach of Sasson and Touchstone (2014). The variables of study and methodology are the same? Is this article a replication of Sasson and Touchstone research? It’s no clear in the manuscript and I consider necessary inform more detailed about it.

a. Authors´ response. We have clarified differences between our study and that of Sasson and Touchstone (2014) (Page 6, lines 124-131).

2. Reviewer´s comment. The section Materials and methods could be called only Method. The Materials or Measures are part of Method.

a. Authors´ response. Following your suggestion, the section ‘Materials and methods’ was renamed as only ‘Method’ (Page 7, line 143).

3. Reviewer´s comment. The subsections Measures might include the instruments used to evaluate the variables of studies. The dependent variables are included in Data analysis. Changed this at measures section.

a. Authors´ response. According to your recommendation, we have added a subsection entitled ‘Measures’ into the section ‘Method’. This subsection comprises information relative to the instruments used to evaluate variables and the dependent variables definition (Page 9, lines 212-213; page 10; and page 11, lines 238-239).

4. Reviewer´s comment. In Table 1, the information about age and gender is redundant because this information is in Participants section. It’s better inserts this table in Results in a new section titled Descriptive characteristics of all variables of the study (including dependent variables) in the two groups, eliminating the section participants of the result.

a. Authors´ response. Following your advice, we have moved Table 1 to the ‘Results’ section, substituted the subsection ‘Participants’ for a new one entitled ‘Descriptive characteristics of variables of the study’, and modified information provided in the table (i.e., we excluded age and gender information due to its redundance and we added descriptive data for dependent variables) (Pages 11-14).

5. Reviewer´s comment. The correlation analysis result it is no clear. I recommend insert a Table with the correlations between variables.

a. Authors´ response. In order to clarify correlation analyses, we have included a new table (entitled ‘Table 2. Spearman correlation between dependent variables and NEPSY-II indexes’) (Pages 17-18). We have also corrected an erratum in the reported statistics, where it said (rs(19) = .775, p = .04), now it appears (rs(7) = .775, p = .04) (Page 17, line 332).

---

## [Decision Letter · Decision Letter 1]

24 May 2021

Visual  preference for social vs. non social images in young children with autism spectrum disorders. An eye tracking study

PONE-D-20-36078R1

Dear Dr. Vacas Ruiz,

You have adequately responded to the comments from the reviewer. We’re pleased to inform you that your manuscript has been judged scientifically suitable for publication and will be formally accepted for publication once it meets all outstanding technical requirements.

Kind regards,

Robert Didden

Academic Editor

PLOS ONE

Additional Editor Comments (optional):

Reviewers' comments:

Reviewer's Responses to Questions

**Comments to the Author**

1. If the authors have adequately addressed your comments raised in a previous round of review and you feel that this manuscript is now acceptable for publication, you may indicate that here to bypass the “Comments to the Author” section, enter your conflict of interest statement in the “Confidential to Editor” section, and submit your "Accept" recommendation.

Reviewer #1: All comments have been addressed

2. Is the manuscript technically sound, and do the data support the conclusions?

Reviewer #1: (No Response)

3. Has the statistical analysis been performed appropriately and rigorously? 

Reviewer #1: (No Response)

4. Have the authors made all data underlying the findings in their manuscript fully available?

Reviewer #1: (No Response)

5. Is the manuscript presented in an intelligible fashion and written in standard English?

Reviewer #1: (No Response)

6. Review Comments to the Author

Reviewer #1: (No Response)

7. PLOS authors have the option to publish the peer review history of their article (what does this mean?). If published, this will include your full peer review and any attached files.

Reviewer #1: No

---

## [Editor Report · Acceptance letter]

26 May 2021

PONE-D-20-36078R1 

Visual preference for social vs. non-social images in young children with autism spectrum disorders. An eye tracking study 

Dear Dr. Vacas:

I'm pleased to inform you that your manuscript has been deemed suitable for publication in PLOS ONE. Congratulations! Your manuscript is now with our production department. 

Kind regards, 

on behalf of

Professor Robert Didden 

Academic Editor

PLOS ONE